# Cerebellum-enriched protein INPP5A contributes to selective neuropathology in mouse model of spinocerebellar ataxias type 17

Qiong Liu[1,2], Shanshan Huang[3], Peng Yin[4], Su Yang[4], Jennifer Zhang[2], Liang Jing[5], Siying Cheng[1,2], Beisha Tang[1,6,7], Xiao-Jiang Li[4], Yongcheng Pan[1,2 ✉] & Shihua Li [4 ✉]

Spinocerebellar ataxias 17 (SCA17) is caused by polyglutamine (polyQ) expansion in the TATA box-binding protein (TBP). The selective neurodegeneration in the cerebellum in SCA17 raises the question of why ubiquitously expressed polyQ proteins can cause neuro-degeneration in distinct brain regions in different polyQ diseases. By expressing mutant *TBP* in different brain regions in adult wild-type mice via stereotaxic injection of adeno-associated virus, we found that adult cerebellar neurons are particularly vulnerable to mutant TBP. In SCA17 knock-in mice, mutant TBP inhibits SP1-mediated gene transcription to down-regulate INPP5A, a protein that is highly abundant in the cerebellum. CRISPR/Cas9-mediated deletion of *Inpp5a* in the cerebellum of wild-type mice leads to Purkinje cell degeneration, and *Inpp5a* overexpression decreases inositol 1,4,5-trisphosphate ($IP_3$) levels and ameliorates Purkinje cell degeneration in SCA17 knock-in mice. Our findings demonstrate the important con-tribution of a tissue-specific protein to the polyQ protein-mediated selective neuropathology.

[1] Key Laboratory of Hunan Province in Neurodegenerative Disorders, Department of Neurology, Xiangya Hospital, Central South University, Changsha, Hunan, China. [2] Department of Human Genetics, Emory University School of Medicine, Atlanta, GA 30322, USA. [3] Department of Neurology, Tongji Hospital, Huazhong University of Science and Technology, Wuhan, China. [4] Guangdong-Hongkong-Macau Institute of CNS Regeneration, Ministry of Education CNS Regeneration Collaborative Joint Laboratory, Jinan University, Guangzhou, China. [5] Department of Emergency, Tongji Hospital, Huazhong University of Science and Technology, Wuhan, China. [6] National Clinical Research Center for Geriatric Disorders, Xiangya Hospital, Central South University, Changsha, Hunan, China. [7] Center for Medical Genetics, School of Life Sciences, Central South University, Changsha, Hunan, China. ✉email: panyongcheng@csu.edu.cn; lishihualis@jnu.edu.cn

Polyglutamine (polyQ) disorders are a group of neurodegenerative diseases caused by a CAG trinucleotide repeat expansion that encodes for an expanded polyQ tract in disease proteins[1]. To date, nine polyQ disorders have been identified: six spinocerebellar ataxias (SCA) types 1, 2, 3, 6, 7, 17; Huntington's disease (HD); dentatorubral–pallidoluysian atrophy; and spinobulbar muscular atrophy[1,2]. An interesting pathological feature of polyQ diseases is that the widely expressed expanded polyQ proteins cause selective neurodegeneration in distinct brain regions in each disease[3]. For example, HD is characterized by preferential loss of the medium spiny neurons in the striatum[4,5], while most SCA diseases show atrophy in the cerebellum and severe loss of Purkinje cells[2,6,7]. The selective neurodegeneration in polyQ diseases resembles the neuronal vulnerability in other age-dependent neurodegenerative diseases such as Alzheimer's and Parkinson's disorders and highlights the importance in understanding its mechanism in order to develop effective treatment for neurodegeneration.

However, despite the monogenic nature of polyQ diseases, understanding the selective neurodegeneration in these diseases has proven to be extremely challenging. This is because the function of most polyQ proteins remains to be defined. Thus investigating polyQ expansion in a well-characterized protein would provide a better opportunity to understand polyQ disease pathogenesis and overcome previous challenges. Spinocerebellar ataxias type 17 (SCA17) is caused by polyQ expansion (>41 glutamines) in the TATA box-binding protein (TBP)[8–11]. TBP is a well-characterized transcription initiation factor that plays a pivotal role in the activation of eukaryotic genes transcribed by RNA polymerase II[12–14]. Similar to other polyQ disease proteins, mutant TBP is expressed ubiquitously throughout the brain and peripheral tissues but preferentially causes neurodegeneration in the cerebellum[8,15]. Given the well-characterized function of TBP and the distinct cerebellar degeneration in SCA17, SCA17 could be an ideal disease model for investigating the mechanism underlying the selective neuropathology, which would also have important implications for understanding the pathogenesis of other SCA diseases.

In the present study, we used SCA17 mice to explore how mutant TBP selectively affects the cerebellum and Purkinje cells. By expressing mutant TBP via adeno-associated viruses (AAV) in different brain regions in wild-type (WT) mice, we found that the cerebellum is the most vulnerable brain region. Using SCA17 knock-in mice that endogenously express mutant TBP, we found that INPP5A, a protein that is highly abundant in the cerebellum and functions as a terminator of the inositol 1,4,5-trisphosphate (IP$_3$) second messenger[16–18], is downregulated. This was caused by the abnormal binding of mutant TBP to the transcription factor SP1, leading to reduced Inpp5a transcription. Furthermore, altering INPP5A in the cerebellum can modulate IP$_3$ levels and cerebellum degeneration in SCA17 knock-in mice. These findings uncover a tissue-specific protein that plays a critical role in the pronounced pathology in the cerebellum and also provide a therapeutic target in SCA diseases.

## Results

**Overexpressed mutant TBP preferentially affects Purkinje cells in the cerebellum**. It is known that polyQ disease neuropathology is dependent on polyQ repeat length, mutant protein expression levels, and cell types. Because expression levels of mutant proteins can vary in different types of cells, whether polyQ-related neuropathology is brain-region-dependent remains to be defined. This issue can be addressed by using stereotaxic injection of adenoviral vector (AAV) expressing the same amounts of mutant TBP in different mouse brain regions, which can avoid the

influence of intrinsically diverse expression levels of mutant TBP in distinct brain regions. To this end, we generated AAV-TBP expressing mutant TBP with different polyQ repeat length (TBP-13Q, -44Q, -68Q, and -105Q) and tested their expression in cultured human embryonic kidney 293 (HEK293) cells (Fig. 1a). Western blotting confirmed that mutant TBP proteins with a series of polyQ repeats were expressed with the expected sizes and at similar levels (Fig. 1b). These viral vectors were packaged into AAV9 viruses and then stereotaxically injected into the cerebellum, striatum, and prefrontal cortex in WT mice (Fig. 1c). Immunofluorescent staining of TBP verified the expression of AAV-TBP in the injected brain areas (Supplementary Fig. 1a).

PolyQ protein toxicity is reflected by its misfolding, which can result in aggregates. We compared mutant TBP aggregates in different brain regions and found that TBP aggregation is increased by a large repeat (105Q) in the prefrontal cortex, striatum, and cerebellum, which is evident by pronounced aggregates formed by TBP-105Q (Fig. 1d–f). However, mutant TBP-68Q appeared to form aggregates only in the cerebellum, but not in the prefrontal cortex and striatum, which was confirmed by quantifying the aggregate density in these brain regions (Fig. 1g). These results suggest that polyQ expansion promotes TBP to preferentially form aggregates in the cerebellum even with smaller repeat lengths.

Next, we wanted to examine whether mutant TBP also differentially affects different types of neuronal cells when same amount of AAV-TBP was used for transduction for the same period length of time. Double immunofluorescent staining clearly showed that mutant TBP-44Q could reduce the number of Purkinje cells, which are labeled by calbindin, in the cerebellum to the same extent as TBP-68Q and TBP-105Q (Fig. 2a). However, only TBP-105Q could drastically diminish the number of NeuN-positive cells in the cerebellum, suggesting that only TBP with a very large polyQ repeat could affect different types of neurons (Fig. 2a). Nissl staining also confirmed that only TBP-105Q decreased the number of granular cells in the cerebellum (Supplementary Fig. 1b, c). In the striatum, TBP-68Q and TBP-105Q dramatically reduced the number of DARPP32-positive cells (Fig. 2b), while mutant TBP with different polyQ repeats slightly reduced the number of NeuN-positive cells in the prefrontal cortex (Fig. 2c). The differential effects of polyQ length-dependent neurotoxicity in brain regions were validated by quantifying the numbers of neurons that were identified by cell-type-specific antibodies (Fig. 2d).

Reactive astrocytes are another hallmark of neurodegenerative diseases and can be assessed by astrocyte marker glial fibrillary acidic protein (GFAP) staining[19–21]. In the cerebellum, compared with TBP-13Q, both TBP-68Q and TBP-105Q caused significantly increased GFAP staining, while TBP-44Q only slightly increased astrocyte reactivity (Supplementary Fig. 2a). Consistent with less severe effects of mutant TBP in the striatum and prefrontal cortex, only a slight increase in reactive astrocytes was seen in these two brain regions (Supplementary Fig. 2b–d). Taken together, our results indicated that the cerebellum is the most vulnerable brain region to mutant TBP toxicity and Purkinje cells are preferentially affected by mutant TBP.

**Selective neuronal loss is also caused by endogenously expressed mutant TBP**. It is important to validate the differential effects of mutant TBP in various brain regions when mutant TBP is expressed at the endogenous level. We therefore examined the previously generated SCA17 knock-in mouse model that endogenously expresses full-length mutant TBP-105Q[22]. TBP-105Q readily forms aggregates in adult SCA17 mice to the extent that western blotting is only able to detect aggregated proteins and

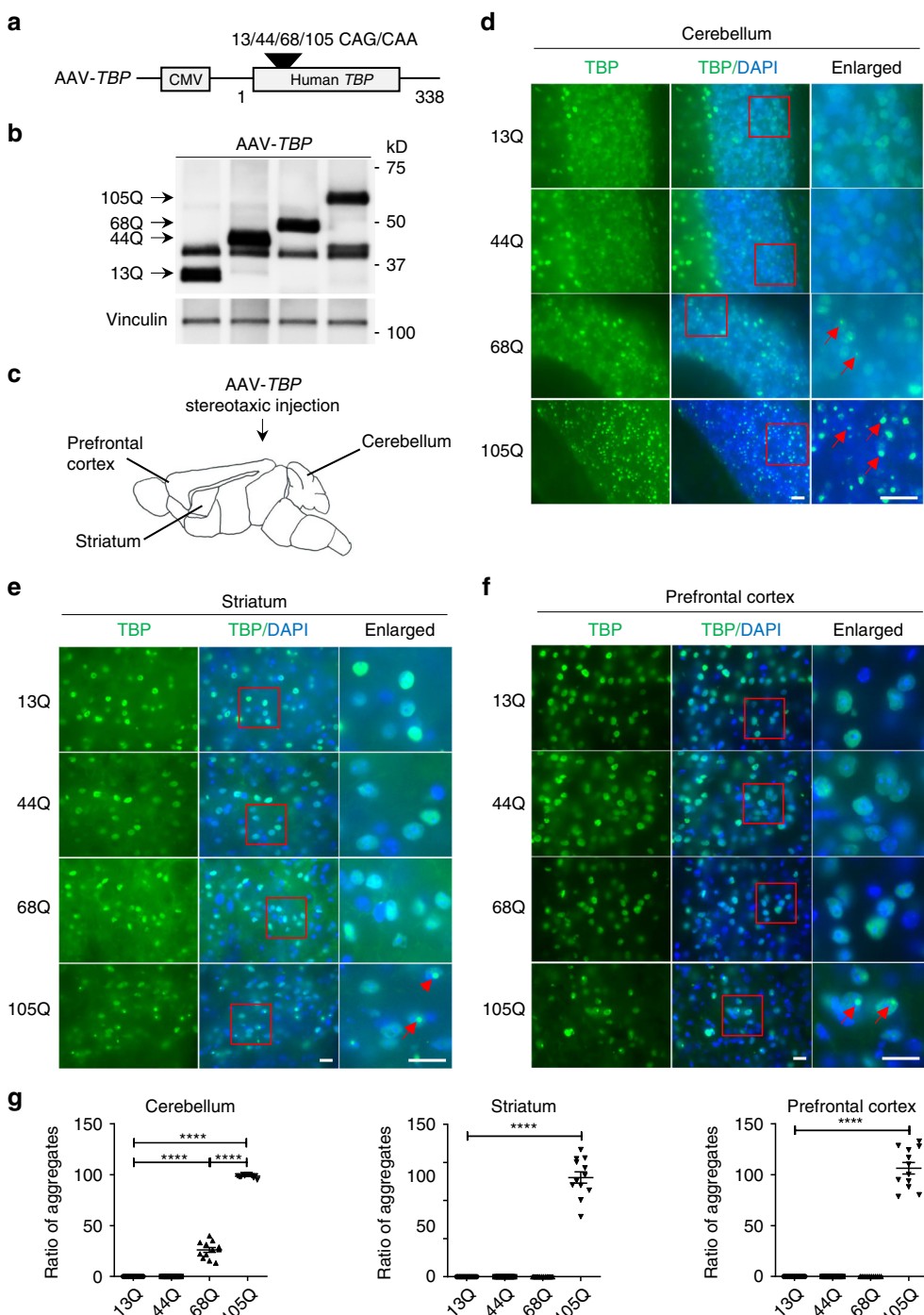

**Fig. 1 PolyQ expansion promotes TBP to preferentially form aggregates in the cerebellum. a** A schematic diagram of AAV plasmids expressing human *TBP* with different polyQ repeat lengths. **b** Western blot analysis of HEK293 cells transfected with AAV-*TBP* plasmids confirming the expression of *TBP* with different polyQ repeats (13Q, 44Q, 68Q, and 105Q). Vinculin was used as a loading control. **c** A diagram of stereotaxic injection of AAV-*TBP* into the cerebellum, striatum, and prefrontal cortex in 3-month-old wild-type mice. **d**–**f** TBP immunofluorescent staining of the cerebellum (**d**), striatum (**e**), and prefrontal cortex (**f**) from AAV-*TBP*-injected mouse brains. Anti-1TBP18 antibody was used. Enlarged images in the boxed areas were also presented. Red arrows indicate aggregates. Scale bar = 20 μm. **g** Quantification of the accumulated TBP aggregates in the cerebellum, striatum, and prefrontal cortex. One-way ANOVA followed with Tukey's multiple comparisons test was performed; cerebellum, $F = 1491$; striatum, $F = 311.6$; prefrontal cortex, $F = 354.6$, ****$P < 0.0001$. $n = 4$ mice per group, three images per brain region from each mouse were used to count. Data are represented as mean ± SEM. Source data and full blots are provided as a Source Data file.

unable to detect soluble mutant TBP in brain tissues[22]. Consistent with our earlier results from AAV-*TBP*-injected mice, western blotting showed that endogenous mutant TBP is more likely to aggregate in the cerebellum than in the striatum and prefrontal cortex in 3-month-old SCA17 knock-in mice (Fig. 3a). We then

compared neuronal vulnerability in different brain regions in SCA17 mice. Because Purkinje cells are specifically localized between the molecular layer and granule layer in the cerebellum, immunofluorescent staining of calbindin-positive cells can determine whether there is loss of Purkinje cells. This staining

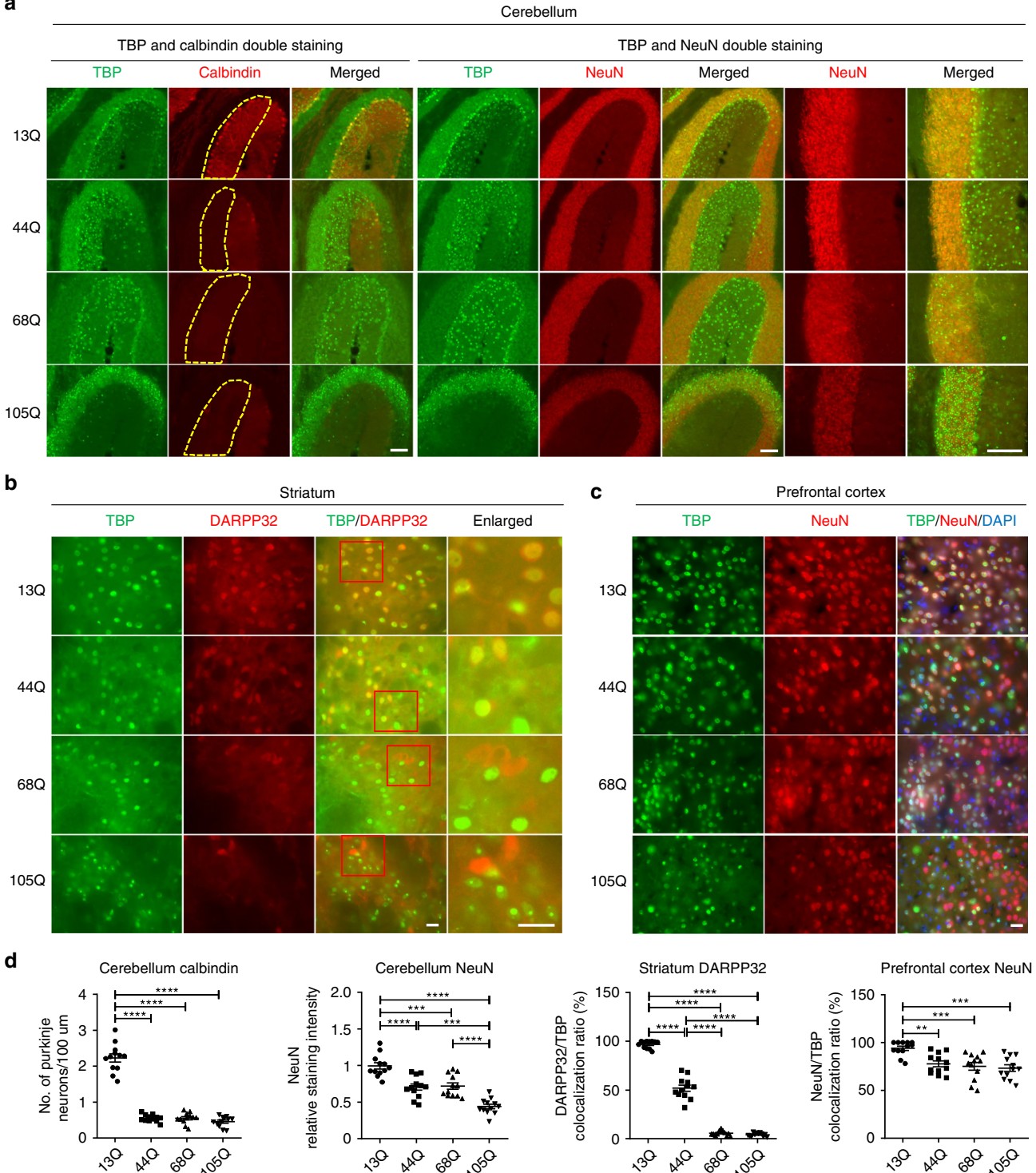

**Fig. 2 Differential effects of mutant TBP in different brain regions. a** Immunofluorescent staining of the cerebellum from AAV-*TBP*-injected wild-type mouse brains. Antibodies to TBP, NeuN, and calbindin were used. Scale bar = 100 μm. **b** Immunofluorescent staining of the striatum from AAV-*TBP*-injected wild-type mouse brain. Staining with antibodies to TBP and DARPP32, a marker for medium spiny neurons, showed a reduced expression of DARPP32 in a polyQ length-dependent manner. Scale bar = 20 μm. **c** Immunofluorescent staining of the prefrontal cortex from AAV-*TBP*-injected wild-type mouse brain with anti-TBP and anti-NeuN antibodies. Scale bar = 20 μm. **d** Quantification of neuronal loss in the cerebellum, striatum, and prefrontal cortex from AAV-*TBP*-injected wild-type mouse brains. One-way ANOVA followed with Tukey's multiple comparisons test was performed; cerebellum calbindin, $F = 166.7$; cerebellum NeuN, $F = 31.65$; striatum, $F = 638.1$; prefrontal cortex, $F = 8.504$, $**P < 0.005$, $***P < 0.0005$, $****P < 0.0001$, $n = 4$ mice per group, three images per brain region from each mouse were used to count. Data are represented as mean ± SEM. Source data are provided as a Source Data file.

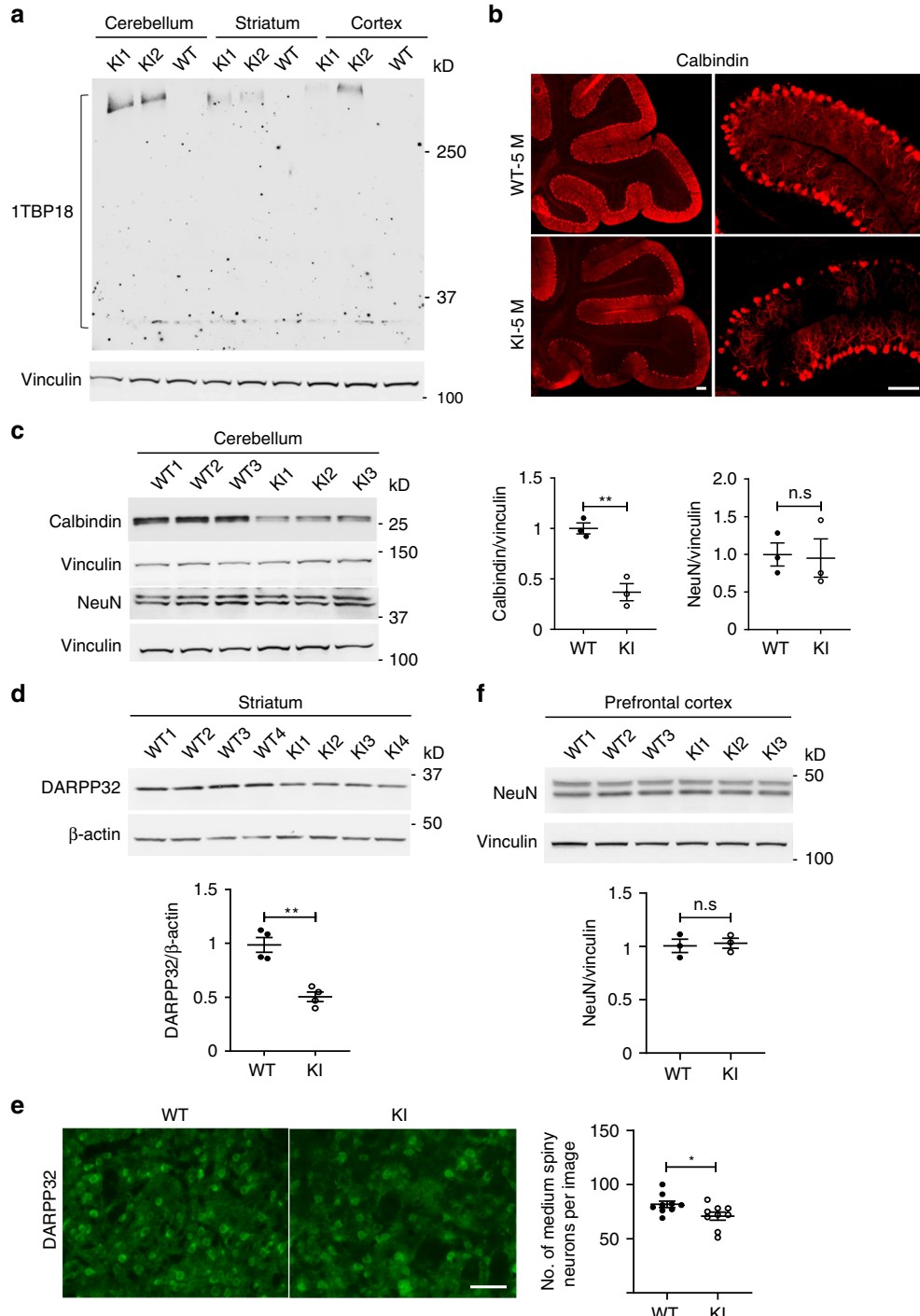

**Fig. 3 Selective neuronal loss is also caused by endogenously expressed mutant TBP. a** Western blotting analysis of TBP expression in the cerebellum, striatum, and prefrontal cortex from 3-month-old SCA17 knock-in (KI) and wild-type (WT) mice. Vinculin was used as a loading control. **b** Immunofluorescent staining of Purkinje neurons with anti-calbindin antibody in the cerebellum of 5-month-old WT and KI mice. Scale bar = 100 μm. **c** Western blotting analysis of calbindin and NeuN levels in the cerebellum of 5-month-old WT and KI mice. The densitometric ratios of calbindin to vinculin and NeuN to vinculin were normalized to WT and analyzed with Student's $t$ test, $t = 6.305$, **$P = 0.0032$, $n = 3$ mice per group. **d** Western blotting of DARPP32 levels in the striatum of 5-month-old WT and KI mice. β-Actin was used as a loading control. The densitometric ratios of DARPP32 to β-actin were normalized to WT and analyzed with Student's $t$ test, $t = 5.872$, **$P = 0.0011$, $n = 4$ mice per group. **e** Immunofluorescent staining with anti-DARPP32 antibody in the striatum of 5-month-old WT and KI mice. Scale bar = 50 μm. Quantitative assessment of the number of medium spiny neurons labeled by DARPP32 per image field are also presented, $t = 2.301$, *$P = 0.0352$, $n = 3$ mice per group, three images from each mouse were used to count. **f** Western blotting of NeuN levels in the prefrontal cortex of 5-month-old WT and KI mice. The densitometric ratios of NeuN to vinculin were normalized to WT and analyzed with Student's $t$ test. $n = 3$ mice per group. Data are represented as mean ± SEM. Source data and full blots are provided as a Source Data file.

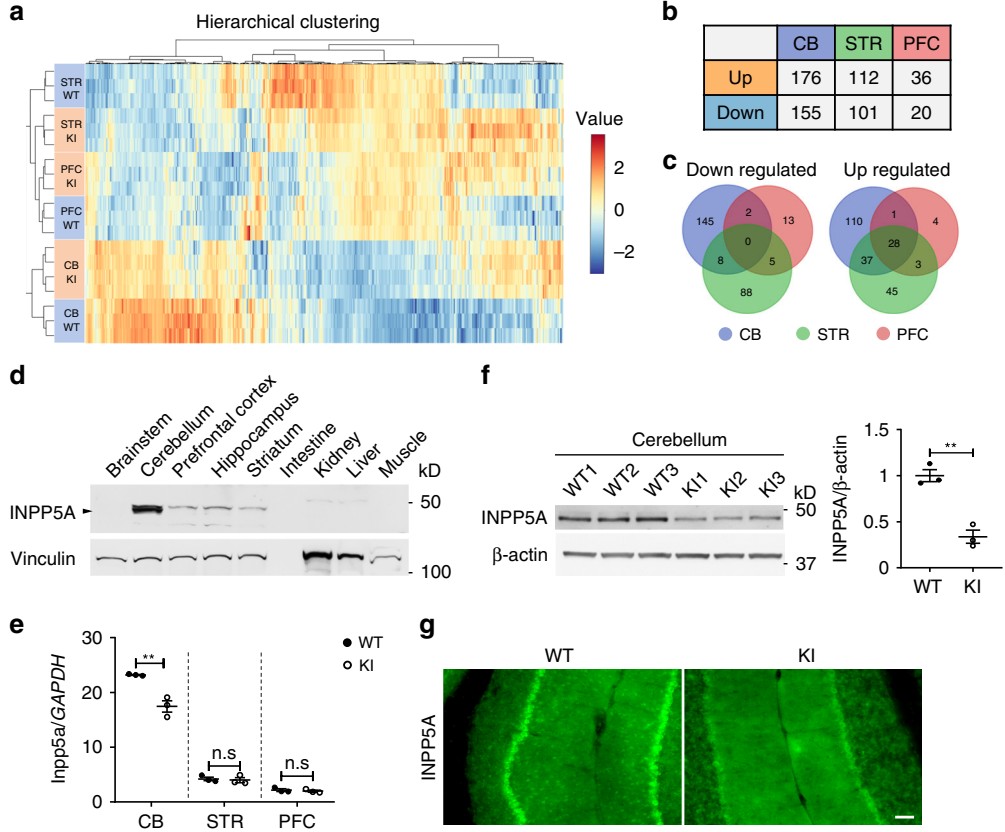

**Fig. 4 Cerebellum-enriched INPP5A is downregulated in SCA17 knock-in mice. a** Heatmaps of dysregulated genes hierarchically clustered from the cerebellum (CB), striatum (STR), and prefrontal cortex (PFC) of 3-month-old SCA17 KI and age-matched WT mice ($n = 3$–4 mice per group). Data were presented as log2 fold change with adjusted $P$ value < 0.05. **b** Summary of the numbers of differentially expressed genes in the CB, STR, and PFC in KI mice. **c** Venn diagrams indicating the numbers of downregulated or upregulated genes in the CB, STR, and PFC. **d** Western blotting of INPP5A levels in different tissues from 3-month-old WT mice. Vinculin was used as a loading control. **e** Real-time PCR assay of *Inpp5a* mRNA levels in the CB, STR, and PFC from 3-month-old KI mice. The relative mRNA levels of *Inpp5a* were obtained by normalizing values to an internal control, *GAPDH*, and analyzed with Student's *t* test, $t = 5.493$, **$P = 0.0054$, $n = 3$ mice per group. **f, g** Western blotting (**f**) and immunofluorescent staining (**g**) of INPP5A in the cerebellum from 3-month-old WT and KI mice. Scale bar = 100 μm. In **f**, densitometric ratios of NeuN to β-actin were normalized to WT and analyzed with Student's *t* test, $t = 6.903$, **$P = 0.0023$, $n = 3$ mice per group. Data are represented as mean ± SEM. Source data and full blots are provided as a Source Data file.

clearly showed a drastic loss of Purkinje cells in SCA17 knock-in cerebellum as compared with WT control (Fig. 3b). We also performed western blotting of calbindin, which allowed for quantitative analysis of the relative level of calbindin, and found a significant reduction (Fig. 3c). However, NeuN expression in the cerebellum was not affected by endogenously expressed mutant TBP (Fig. 3c). In the striatum, reduced DARPP32 protein level was also seen in the striatum of 5-month-old SCA17 knock-in mice (Fig. 3d). Moreover, immunofluorescent staining showed a slightly but significantly reduced number of DARPP32-positive cells (Fig. 3e). However, NeuN expression in the prefrontal cortex was not changed (Fig. 3f). These results suggest that mutant TBP at the endogenous level also preferentially affects Purkinje cells and striatal cells, which is consistent with the selective loss of Purkinje cells and degeneration of striatal neuronal cells in SCA17 patients[8,15].

**Cerebellum-enriched INPP5A is downregulated in SCA17 knock-in mice.** The above results suggest that the most prominent neuronal degeneration in SCA17 knock-in mice occurs in the cerebellum. Given that TBP is a ubiquitously expressed transcription factor, the selective neurotoxicity of mutant TBP could involve the alteration in the expression or function of tissue-specific proteins. To identify such targets, RNA sequencing was performed using cerebellum, striatum, and prefrontal cortex

tissues from 3-month-old SCA17 knock-in mice and WT littermates (Fig. 4a). As expected, the cerebellum from SCA17 knock-in mice displayed the greatest number of differentially expressed genes, while the prefrontal cortex had the least number of affected genes (Fig. 4b, Supplementary Data 1). Gene set enrichment analyses were performed on the differentially expressed genes in different brain regions (Supplementary Fig. 3). Interestingly, the 28 genes that were upregulated in all three brain regions (Fig. 4c) were enriched in immune-related pathways (Supplementary Fig. 3g), which is consistent with increased reactive astrogliosis in AAV-*TBP*-injected mouse brains (Supplementary Fig. 2).

RNA sequencing identified 245 cerebellar specifically dysregulated genes (110 upregulated genes and 145 downregulated genes). However, all of the 110 upregulated genes were expressed at the comparable level in three brain regions and are involved in immune-related pathway (Supplementary Data 1, Supplementary Fig. 3a). Therefore, we focused on the 145 specifically downregulated genes and ranked them based on their abundance and specificity in the cerebellum (Supplementary Data 1). Among these genes, *Inpp5a* is an interesting candidate gene because of its selective expression in the cerebellum. INPP5A protein is the major enzyme that hydrolyzes $IP_3$, an intracellular messenger that increases intracellular calcium to mediate cell responses to various stimulations[16,23,24]. *Inpp5a* deletion was also reported to cause ataxia in mice[25]. The *Inpp5a* gene has three different

splicing isoforms, named A, B, and C, encoding predicted proteins of 412aa, 422aa, and 420aa, respectively (Supplementary Fig. 4a). Sequence alignment reveals that isoform A has a distinct C terminus, whereas isoform C has a unique N terminus. PCR studies using isoform-specific primers showed that isoforms (A and C) are highly expressed in the cerebellum compared to the prefrontal cortex. However, isoform C is expressed at a very low level in both the cerebellum and cortex (Supplementary Fig. 4b, c), whereas isoform A is much more abundant in the cerebellum than in the cortex. Western blotting also revealed that, among all the tissues examined, *Inpp5a* is expressed at a much higher level in the cerebellum (Fig. 4d).

RNA sequencing results showed that *Inpp5a*, but not other *Inpp5* paralogs (Supplementary Fig. 4d), was significantly decreased in the SCA17 knock-in mouse cerebellum compared to the WT littermates. Real-time PCR assay verified the decreased level of *Inpp5a* transcripts in the cerebellum but not in the striatum and prefrontal cortex in SCA17 knock-in mice (Fig. 4e). Moreover, western blotting results showed that INPP5A was significantly reduced in the cerebellum of SCA17 knock-in mice (Fig. 4f). Immunofluorescent staining demonstrated that INPP5A is highly expressed in Purkinje cells in WT mice but is markedly decreased in SCA17 knock-in mice (Fig. 4g). These results suggested that mutant TBP reduces the expression of *Inpp5a* mRNA, leading to a decrease in its protein level.

**Expanded polyQ affects the interaction between TBP and SP1 and impairs the transcription of *Inpp5a*.** Given that *Inpp5a* is highly expressed in Purkinje cells and its deletion is associated with ataxia phenotypes in mice[25], we then explored how mutant TBP can affect *Inpp5a* transcription. Since there is no TATA box sequence present in the *Inpp5a* promoter proximal region, mutant TBP may not directly bind to its promoter to inhibit its transcription. However, we found several sites that can bind zinc finger transcription factor SP1, which showed higher scores for the potential to regulate *Inpp5a* (Supplementary Table 1).

Mutant TBP has been reported to bind to SP1 to affect gene expression in cultured cells[26]. We performed chromatin immunoprecipitation (ChIP) in Neuro2a cells, which had been transfected with *SP1* and *TBP* (either 13Q or 105Q) plasmids, and investigated whether the binding efficiency of SP1 to the two *Inpp5a* isoform (A and C) promoters was affected by mutant TBP. In the presence of mutant TBP, there was a significant reduction in the association of SP1 with the endogenous *Inpp5a* isoform A promoter compared with the control proliferating cell nuclear antigen (*PCNA*) promoter. However, there was a slight but not significant decreased association with *Inpp5a* isoform C promoter (Fig. 5a). Since there was only one SP1-binding site in the remote region of *Inpp5a* isoform C promoter and the expression of isoform C was low (Supplementary Fig. 4b, c), we then focused on isoform A and refer to *Inpp5a* isoform A as *Inpp5a* unless otherwise stated.

To functionally assess the effect of mutant TBP on the transcription of *Inpp5a*, we generated a luciferase reporter vector (Fig. 5b) to express luciferase under the control of the proximal promoter of *Inpp5a* ($-1419$ to $-109$). *TBP* stably transfected PC12 cell lines (13Q or 105Q) were transfected with *SP1* or green fluorescent protein (GFP) plasmid (Fig. 5c). As expected, overexpression of *SP1* dramatically enhanced luciferase intensity, indicating that SP1 indeed activates *Inpp5a* transcription (Fig. 5d). Moreover, *TBP*-105Q cell lines yielded a much lower level of transcriptional activity compared to *TBP*-13Q cells (Fig. 5d). Furthermore, transiently transfected HEK293 cells expressing the *Inpp5a* promoter reporter and *SP1* plus *TBP*-13Q or *TBP*-105Q confirmed the decreased activity of this reporter in the presence

of mutant TBP-105Q (Fig. 5e, f). In addition, *TBP*-44Q and *TBP*-68Q also significantly affected the SP1-mediated *Inpp5a* transcriptional activity (Supplementary Fig. 5).

We then wanted to know whether mutant TBP bound more tightly to SP1 to affect *Inpp5a* expression. To this end, HEK293 cells were transfected with *SP1* and *TBP*-13Q or *TBP*-105Q. Co-immunoprecipitation showed that significantly more TBP-105Q than TBP-13Q was coprecipitated with SP1 (Fig. 5g), suggesting that polyQ expansion enhances the binding of SP1 with mutant TBP and may impair SP1 binding to the *Inpp5a* promoter region, leading to the reduction of *Inpp5a* transcription. To provide in vivo evidence for this idea, immunoprecipitation using anti-TBP antibody was performed with cerebellar extracts from WT and SCA17 knock-in mice. More SP1 was precipitated from SCA17 knock-in mouse brain compared to the brain lysates from WT mice (Fig. 5h). Taken together, the above results suggest that SP1 is involved in transcriptional regulation of *Inpp5a* and mutant TBP could inhibit the transcriptional activity of SP1 through its abnormal interaction.

**Knockdown of *Inpp5a* leads to loss of Purkinje cells while overexpression of *Inpp5a* could ameliorate Purkinje cell degeneration.** Previous studies have shown that *Inpp5a* deletion generated by gene-trap insertion causes perinatal lethality and cerebellar degeneration in a mouse model with mixed genetic background[25]. However, earlier literature indicates that gene-trap insertion for generating gene deletion based on mixed genetic background would affect phenotypes[27]. Thus whether *Inpp5a* knockdown in adult mouse brain could induce Purkinje cell degeneration is still inconclusive. We used CRISPR/Cas9 editing method to knockdown *Inpp5a* in adult mouse brain by stereotaxic injection of AAV viruses carrying *Inpp5a* guide RNAs (gRNAs) into the cerebellum of *Cas9* transgenic mice (Fig. 6a). *Cas9* transgenic mice were generated by crossing EIIa-Cre mice with conditional *Cas9* mice that were obtained from the Jackson Laboratory and described previously[28]. The efficiency of gRNAs targeting *Inpp5a* was verified by T7 Endonuclease I assay before virus packaging (Supplementary Fig. 6a). *Inpp5a* gRNA or control gRNA virus was bilaterally injected into the cerebellum of *Cas9* transgenic mice. After 4 weeks, brain tissues were isolated for analysis. Western blotting showed that the level of INPP5A was dramatically reduced in the *Inpp5a* gRNA-injected side compared to the control side injected with control viruses (Fig. 6b). Since AAV vector also expressed red fluorescent protein (RFP), we performed double immunofluorescent staining of the injected mouse cerebellum. Knockdown of *Inpp5a* led to an obvious reduction in calbindin-labeled Purkinje cell density (Fig. 6c, Supplementary Fig. 6b).

Since *Inpp5a* knockdown in the cerebellum reduced Purkinje cell numbers, it is possible that overexpression of *Inpp5a* could alleviate Purkinje cell degeneration. To test this idea, we generated an AAV-*Inpp5a* expression vector to overexpress mouse *Inpp5a* under the control of a CMV promoter (Fig. 6a). We then injected AAV-*Inpp5a* or control AAV-GFP bilaterally into the cerebellum of 4-month-old SCA17 knock-in mice. One month after injection, brain tissues were collected and the expression of *Inpp5a* or GFP was confirmed by western blotting (Fig. 6d) and immunofluorescent staining (Fig. 6e). Overexpression of *Inpp5a* in SCA17 knock-in mouse brain increased the number of calbindin-labeled Purkinje cells, while AAV-GFP-injected SCA17 knock-in cerebellum still showed a significant reduction of Purkinje cells (Fig. 6e), which was further confirmed by Nissl staining (Supplementary Fig. 6c). Western blotting analysis also showed the increased level of calbindin in the AAV-*Inpp5a*-injected side (Fig. 6f). Quantitative analysis of calbindin-

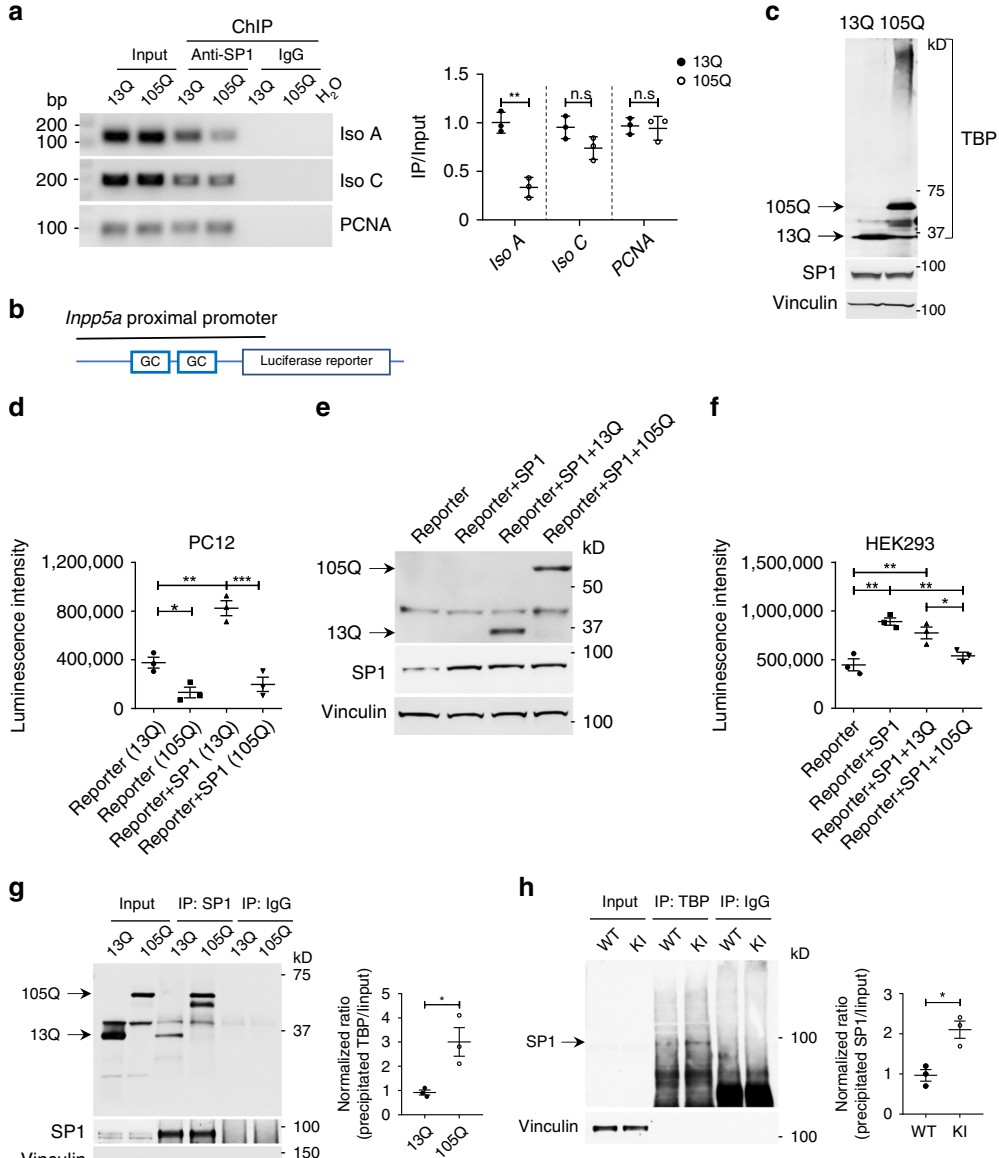

**Fig. 5 Expanded polyQ affects the interaction between TBP and SP1 and impairs the transcription of *Inpp5a*. a** Chromatin immunoprecipitation (ChIP) assay of SP1 association with the *Inpp5a* promoter in HEK293 cells transfected with *TBP*-13Q or *TBP*-105Q. Cross-linked chromatin materials were immunoprecipitated (IP) by anti-SP1 and were subjected to polymerase chain reaction (PCR) with primers specific for the promoter region of *Inpp5a* and *PCNA* (proliferating cell nuclear antigen), which served as a control. Representative agarose gel electrophoresis images of PCR products (left). Quantification of the ratios of immunoprecipitated products to input is presented (right). $T = 7.757$, $**P = 0.0015$ (unpaired $t$ test). **b** Schematic map of a luciferase construct for reporter assay. **c** Western blotting showing the expression of SP1 and TBP in transfected PC12 cells. **d** Luciferase activity analysis of stable PC12 cell lines expressing *TBP*-13Q or *TBP*-105Q with or without transfected *SP1*. Luminescence intensity was analyzed with one-way ANOVA followed with Tukey's multiple comparisons test. $F = 35.19$, $*P < 0.05$, $**P < 0.005$, $***P < 0.0005$. **e** Western blot analysis of HEK293 cells transfected with *SP1* and *TBP*-13Q or 105Q. **f** Luciferase activity analysis of transfected HEK293 cells. Cells transfected only with *Inpp5a* promoter reporter were used as control. Luminescence intensity was analyzed with one-way ANOVA followed with Tukey's multiple comparisons test. $F = 16.39$, $*P < 0.05$, $**P < 0.005$. **g** Co-immunoprecipitation of transfected *SP1* with *TBP*-13Q or *TBP*-105Q in HEK293 cells showing an increased interaction of SP1 with soluble TBP-105Q compared to TBP-13Q. Right panel showing the ratio of precipitated TBP to input. $T = 3.518$, $*P = 0.0245$ (Student's $t$ test). **h** TBP antibody immunoprecipitation of mouse cerebellar lysates also showed that more SP1 was coprecipitated in KI mice than in WT mice. One-month-old mice were used. Right panel showing the ratio of precipitated SP1 to input. $T = 4.389$, $*P = 0.0118$ (Student's $t$ test). Data are represented as mean ± SEM. Source data and full blots are provided as a Source Data file.

positive cells in immunofluorescent staining and the relative expression level of calbindin in western blots also verified that INPP5A is protective for Purkinje cells (Fig. 6g). All these results confirmed the protective effect of INPP5A on mutant TBP toxicity and its involvement in the Purkinje neuron degeneration.

Previous studies demonstrated the critical role of INPP5A in regulating IP₃ and intracellular calcium by terminating the IP₃

signal[16,29,30]. Perturbation of IP₃ signal has been implicated in the pathogenesis of SCAs[16,31–35]. Therefore, we detected the IP₃ level using IP₃ enzyme-linked immunosorbent assay (ELISA) assay. We found that the IP₃ level was significantly increased in adult SCA17 knock-in cerebellar lysate (Fig. 6h, left). We further examined the IP₃ level after AAV-*Inpp5a* or AAV-GFP injection into the cerebellum of adult SCA17 knock-in mice and found that

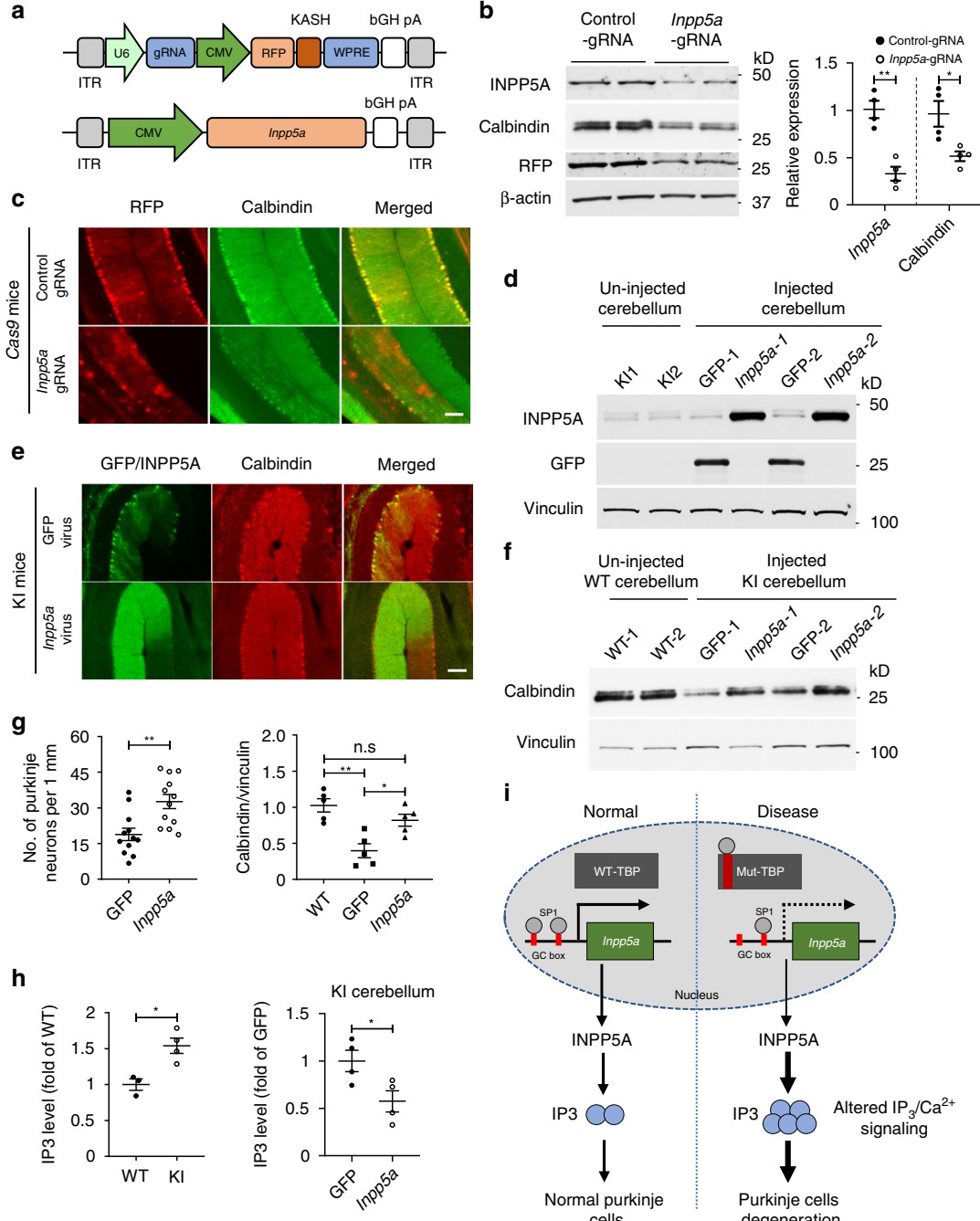

**Fig. 6 *Inpp5a* Knockdown leads to Purkinje cell degeneration, whereas *Inpp5a* overexpression reduces Purkinje cell degeneration. a** Schematic diagram of AAV-*Inpp5a* gRNA and AAV-*Inpp5a* vectors. **b** Western blotting analysis of INPP5A and calbindin in AAV-control gRNA or AAV-*Inpp5a* gRNA-injected 3-month-old EIIa-*Cas9* transgenic mice cerebellum (left panel). The densitometric ratios of INPP5A to β-actin and calbindin to β-actin were normalized to control and analyzed with Student's $t$ test (right panel). Calbindin, $t = 3.095$, $^*P = 0.0213$; *Inpp5a*, $t = 5.79$, $^{**}P = 0.0012$, $n = 4$ mice per group.
**c** Representative immunofluorescence images of AAV-gRNA-injected *Cas9* mouse cerebellum without mutant TBP. **d** Western blotting showing the expression of AAV-*Inpp5a* or AAV-GFP in the injected KI mouse cerebellum. **e** Representative immunofluorescence images of AAV-*Inpp5a* or AAV-GFP-injected 5-month-old KI mouse cerebellum. **f** Western blotting showing increased calbindin in the cerebellum of SCA17 mice after injection of AAV-*Inpp5a*. AAV-GFP injection served as control. **g** Quantification of Purkinje cells (left panel) in **e** was analyzed with Student's $t$ test, $t = 3.483$, $^*P = 0.0011$, $n = 4$ mice per group, three images were used to count from each mouse. The densitometric ratios of calbindin to vinculin in **f** were normalized to WT and analyzed with one-way ANOVA followed with Tukey's multiple comparisons test. $F = 12.32$, $^*P < 0.05$, $^{**}P < 0.005$. $n = 5$ mice per group. **h** IP3 ELISA assay of cerebellar lysates from 2-month-old KI mice and age-matched WT mice (left) and from AAV-*Inpp5a* or AAV-GFP-injected 3-month-old KI mice (right). IP3 levels (fold of WT or fold of GFP) were analyzed with Student's $t$ test, $t = 3.77$, $^*P = 0.013$ (left); $t = 2.681$, $^*P = 0.0365$ (right), $n = 3$–4 mice per group. **i** A proposed model for selective neurodegeneration in SCA17. Mutant TBP downregulates INPP5A by inhibiting SP1 from activating *Inpp5a* transcription. The reduction of INPP5A dysregulates the IP3/Ca$^{2+}$ pathway, leading to Purkinje cell degeneration. Scale bar = 100 μm. Data are represented as mean ± SEM. Source data and full blots are provided as a Source Data file.

*Inpp5a* overexpression reduced IP$_3$ level compared with the GFP-injected side (Fig. 6h, right). In combination with the protective effect of INPP5A against Purkinje cell degeneration, we propose that INPP5A plays a critical role in selective neurodegeneration in SCA17 because mutant TBP impairs its expression to affect IP$_3$/Ca$^{2+}$ signal pathway, leading to Purkinje cell degeneration (Fig. 6i).

## Discussion

As an important transcription factor, TBP is essential for the function and viability of a vast majority of cells. However, how polyQ expansion in this ubiquitously expressed transcription factor can cause selective cerebellar neuronal degeneration is still unclear. In the current study, we provide strong evidence that this selective vulnerability involves *Inpp5a* that is highly expressed in Purkinje cells. Our findings highlight the importance of cell-type-specific proteins for the selective neurodegeneration in polyQ diseases.

Several lines of evidence in our findings support the critical role of INPP5A in SCA17 pathogenesis. First, when mutant *TBP* is overexpressed via AAV vector transduction or expressed at the endogenous level in SCA17 knock-in mice, it preferentially affects Purkinje cells, suggesting that Purkinje cell specific protein(s) may be involved in this selective vulnerability. Second, western blotting and immunofluorescence results indicated that *Inpp5a* is highly expressed in the cerebellum, especially in Purkinje cells. Third, INPP5A is reduced in SCA17 knock-in cerebellum, and more importantly, mutant TBP binds SP1 to inhibit *Inpp5a* transcription. Finally, knockdown of *Inpp5a* in the cerebellum of WT mice led to Purkinje cell degeneration, whereas overexpression of *Inpp5a* in the cerebellum of SCA17 knock-in mice alleviated Purkinje cell degeneration.

INPP5A belongs to the inositol polyphosphate 5-phosphatase family of enzymes that inactivate the second messenger molecules IP$_3$ in a signal terminating reaction[16]. IP$_3$ binds to IP$_3$R and stimulates the transient release of Ca$^{2+}$ from the endoplasmic reticulum[16,18,36]. It is well documented that subtle changes in IP$_3$/Ca$^{2+}$ signaling pathway causes abnormal alteration in intracellular Ca$^{2+}$, which contributes to the onset of a range of human diseases, such as Alzheimer's disease, amyotrophic lateral sclerosis, and HD[16,32], and some SCAs, like SCA2 and SCA3[34,37]. In our SCA17 KI mice, cerebellar IP$_3$ level is increased corresponding to the reduced INPP5A, while overexpression of *Inpp5a* significantly decreased IP$_3$ levels in the cerebellum. Moreover, overexpression of *Inpp5a* alleviated Purkinje cell degeneration in SCA17 mice. These results indicated that dysregulated IP$_3$ is involved in mutant TBP-induced selective cerebellar neurodegeneration.

Most SCA diseases share the common pathological feature in Purkinje cells, but the underlying mechanism is still not clear. Kasumu et al. reported that AAV-mediated expression of *Inpp5a* could be used to prevent the onset of Purkinje cell dysfunction and reduce Purkinje cell degeneration in SCA2-58Q mice[38]. In our study, we provide direct evidence for the vital role of INPP5A in the pathogenesis of SCA17, as knockdown of *Inpp5a* in the cerebellum led to Purkinje cell degeneration but overexpression of *Inpp5a* alleviated Purkinje cell degeneration. Since we also compared mutant TBP-mediated neuropathology in different brain regions and since INPP5A is enriched in the Purkinje cells, our finding revealed its selective function in Purkinje cells, highlighting the importance of IP$_3$/Ca$^{2+}$ signaling pathway for SCAs.

In addition, the unique ability of Purkinje cells to deal with misfolded polyQ proteins also contributes to selective vulnerability. By overexpressing mutant *TBP* in different brain regions, we found that TBP-68Q can readily form aggregates in Purkinje

cells. Aggregated mutant TBP is unlikely to be as effective as soluble mutant TBP to bind transcription factors and to affect gene expression. Evidence supporting this idea includes the fact that overexpressed *TBP*-44Q, which does not form obvious aggregates in the cerebellum, still caused Purkinje cell degeneration to the similar extent as overexpressed *TBP*-68Q and *TBP*-105Q. Also, biochemical studies revealed that soluble mutant TBP binds SP1 and reduces *Inpp5a* transcription. However, the extent of aggregated polyQ proteins reflects the accumulation of misfolded proteins, which is a prerequisite for the toxicity of mutant proteins, especially when expressed at the endogenous level.

Another interesting finding in our study is the striatal pathology in SCA17. Overexpression of mutant *TBP* (*TBP*-68Q and *TBP*-105Q) via AAV vector transduction reduced the number of DARPP32-positive cells. Meanwhile, both DARPP32 protein level and DARPP32-positive cells were also reduced in the striatum of 5-month-old SCA17 knock-in mice. The reduced number of DARPP32-positive neurons is consistent with the degeneration of striatal neuronal cells in SCA17 patients. Gene set enrichment analysis on striatum-specific differentiated genes revealed that most downregulated genes are involved in calcium transport or calcium transmembrane transport (Supplementary Fig. 3d, k) or associated with striatal degeneration, mental retardation, dyskinesia, ataxia, and epilepsy (such as *Camk2b*, *Pde10a*, *Kcnab1*, *Tomm70a*, and *Cacna2d3*; Supplementary Data 1). These findings may provide some clues for why SCA17 shares striatal neuropathology and symptoms with HD[39]. However, whether the striatal pathology in SCA17 is mediated by the same mechanism underlying HD neuropathology remains to be investigated.

In summary, our studies identified INPP5A as a cell-type-specific protein that is highly abundant in Purkinje cells and is involved in the selective pathogenesis of SCA17. This finding also underscores the potential importance of other tissue-specific proteins in the selective degeneration of distinct populations of neuronal cells in polyQ diseases. Also, given that altered IP$_3$/calcium signaling is known to associate with ataxia, the identification of INPP5A as the primary molecule involved in the selective neurodegeneration in SCA17 provides a promising therapeutic target for the treatment of SCA17 and perhaps other diseases with ataxia as well.

## Methods

**Animals.** All mice were maintained on a 12:12 h light/dark cycle (lights off at 7 p.m.). The temperature was maintained at 22 ± 1 °C with relative humidity (30–70%). All animal procedures were performed in accordance with the NIH and U.S. Public Health Service's Guide for the Care and Use of Laboratory Animals and were approved by the Institutional Animal Care and Use Committee at Emory University, which is accredited by the American Association for Accreditation of Laboratory Care. The Tbp-105Q germline knock-in mice were generated as described previously[22]. In brief, heterozygous TBP floxed mice were crossed with EIIa-Cre transgenic mice and the resulting knock-in progeny was then crossed with WT mice to obtain germline knock-in mice. Genotyping was performed by PCR and agarose gel electrophoresis with *TBP* primers, Forward: 5'-CCA CAG CCT ATT CAG AAC ACC-3' and Reverse: 5'-AGA AGC TGG TGT GGC AGG AGT-3'.

**Plasmids and viruses.** The *Inpp5a* reporter plasmid was generated by cloning the mouse *Inpp5a* promoter containing SP1 motif sequence GGGCGG from mouse genomic DNA into the pGL4.14 vector using Kpn I and Hind III restriction sites with primers *Inpp5a* reporter primers, Forward: 5'-CCA GAA CCC ATG CTA TAA GAA GAG-3' and Reverse: 5'-AAT CTA CAG ACT TGG GTC TTA GGC-3'. The pRK5-*SP1* plasmid and AAV-CMV-*TBP*-13Q/44Q/68Q/105Q plasmids were generated in our previous studies[40,41]. AAV-CMV-*Inpp5a* plasmid was generated by cloning cDNA of mouse *Inpp5a* isoform A from mouse first-strand cDNA into the AAV-CMV vector using EcoR I and Xba I restriction sites with *Inpp5a* isoform A primers, Forward: 5'-CCG GAA TTC ATG GCG GGG AAG GCG GCC GC-3', and Reverse: 5'-TGC TCT AGA TTA GGA GGA TGA GTT GGA TAC ACT GC-3'. AAV-*Inpp5a*-gRNAs were generated by inserting gRNA into a modified PX552 vector[42] via SapI restriction site. gRNA sequences are: *Inpp5a* gRNA-1: ATT TGG AGG GAA AAA CTA CG agg and *Inpp5a* gRNA-2: TTT GTC CAC ATG GGA

CAT GG agg and control gRNA: ACC GGA AGA GCG ACC TCT TCT (PAM sequences are shown in lowercase; control gRNA was verified previously)[42]. All constructs were confirmed by double direction Sanger sequencing. AAV serotype 9 viruses, including AAV-TBP-13Q/44Q/68Q/105Q, AAV-Inpp5a, AAV-GFP, and AAV-gRNAs viruses were produced at the Viral Vector Core at Emory University. The genomic titers of viruses were determined by PCR.

**Antibodies**. Primary antibodies used in this study include: 1TBP18 (QED Bioscience, 70102, 1:1000), Calbindin (Millipore, ab1778, 1:1000), DARPP32 (Abcam, ab40801, 1:1000), NeuN (Cell Signaling Technology, 24307s, 1:2000), GFP (BD Living Colors, 632376, 1:1000), RFP (Abcam, ab62341, 1:1000), vinculin (Cell Signaling Technology, 13901, 1:5000), β-actin (Cell Signaling Technology, 4967L, 1:10,000), and INPP5A (Invitrogen, PA5-45906, 1:1000), SP1 (Santa Cruz, sc-17824 X, 1:1000). Secondary antibodies were donkey anti-rabbit and donkey anti-mouse Alexa Fluor 488 or 594 from Jackson ImmunoResearch.

**Stereotaxic injection of AAVs into mouse brain**. Stereotaxic surgery was performed as described previously[43]. Briefly, the mice were anesthetized with 1.5% isoflurane inhalation and stabilized on a stereotaxic instrument (David Kopf Instruments). The location for cerebellum injection was determined according to the distance from bregma: anterior–posterior = −6.3 mm, medial–lateral = ±1.7 mm, dorsal–ventral = −1.5 mm. For striatum: anterior–posterior = 0.55 mm, medial–lateral = ±2 mm, dorsal–ventral = −3.5 mm. For prefrontal cortex: anterior–posterior = 2 mm, medial–lateral = ± 0.25 mm, dorsal–ventral = −1.8 mm. Small holes were drilled in the skull, and a 30-gauge Hamilton microsyringe connected to a syringe infusion pump (World Precision Instruments, USA) was used to deliver the viruses at a speed of 200 nl/min. AAVs were injected unilaterally or bilaterally into the brain regions as indicated. Meloxicam at 5 mg/kg was given before surgery as an analgesic, and the mice were placed on a heated blanket to recover from the anesthetic after surgery. Four to five mice per group were used for stereotaxic injection.

**Immunofluorescent staining**. Mice were anesthetized and perfused intracardially with 0.9% saline solution, followed by 4% paraformaldehyde in 0.1 M phosphate buffer at pH 7.2. Brains were removed, cryoprotected in 30% sucrose at 4 °C, and sectioned at 40 μm for subsequent immunofluorescence study: free-floating sections were permeabilized, blocked, and incubated with primary antibodies at 4 °C overnight. After incubation with Alexa Fluor-conjugated secondary antibodies (1:200) and 4′,6-diamidino-2-phenylindole (1:5000), the brain sections were mounted to coated glass slides and visualized with a Zeiss scope.

**Reverse transcriptase PCR and quantitative real-time PCR**. Total RNA was extracted from mice brain tissue using Trizol reagent (Invitrogen) according to the manufacturer's protocol. Complementary DNA (cDNA) was synthesized from total RNA (2 μg) using a High Capacity cDNA Reverse Transcription Kit (4368814, Applied Biosystems, Thermo Scientific) and oligo dT primers. For reverse transcriptase PCR, dNTP, rTaq with matched reaction buffer from TaKaRa were used. Quantitative PCR was carried out on a Master Cycler RealPlex4 System (Eppendorf) using Power SYBR Green PCR Master Mix (4367659, Applied Biosystems, Thermo Scientific) and analyzed with the comparative cycle threshold method (also referred to as the $2^{-\Delta\Delta CT}$ method). Primers for each gene are as follows: Inpp5a primers, Forward: 5′-ACA CAC ACA AGC CTC ACT TCAT-3′ and Reverse: 5′-TCA TTG CGT CAC TGG ATA GTA GTT-3′; GAPDH primers, Forward: 5′-CAT GGC CTC CAA GGA GTA AGA AAC-3′ and Reverse: 5′-TGT GAG GGA GAT GCT CAG TG-3′.

**RNA sequencing and gene enrichment analysis**. Total RNA was extracted from the cerebellum, prefrontal cortex, and striatum of 3-month-old germline SCA17 knock-in mice and age-matched WT mice (three to four mice per genotype) using Trizol reagent (Life Technologies) following the manufacturer's instructions. Messenger RNA sequencing was performed at Emory Yerkes Gen Core using the Illumina HiSeq1000 system. After quality control, reads were mapped and normalized using multiple tools including the Cufflinks software package. Normalized read counts were compared across samples and adjusted for multiple testing using the Cuffdiff package. The resulting P values were adjusted using the Benjamini–Hochberg False Discovery Rate method. Genes with an adjusted P value < 0.05 (in green background) were assigned as differentially expressed. Gene set enrichment analyses of differentially expressed genes were performed in terms of their enrichment by MetaCore's ontologies, including ontologies and Gene Ontology (GO) ontologies: Pathway Maps and GO Processes. To identify cerebellum-specific genes, all genes are listed in order of Reads Per Kilobase per Million mapped reads in WT. Genes whose expression in the cerebellum is at least three times higher than that in the striatum and prefrontal cortex were defined as cerebellum-enriched genes.

**Cell culture and transfection**. HEK293 cells and Neuro2a cells were grown at 37 °C under 5% CO$_2$ in Dulbecco's modified Eagle medium supplemented with 10% fetal bovine serum and 100 U/ml of penicillin/streptomycin. The stable TBP-transfected PC12-expressing cell lines were generated in our previous study[26,41] and were cultured in Dulbecco's modified Eagle medium supplemented with 10% horse serum, 5% fetal bovine serum, 100 U/ml of penicillin/streptomycin, and 800 μg/ml G418. Plasmid transient transfection study was performed using Lipofectamine 2000 reagent (Invitrogen) according to the manufacturer's protocol.

**Luciferase promoter activity assay**. The transcription factor-binding sites analysis was performed on the hypothetical promoter region of Inpp5a (from −2000 to 0) using an online transcription factor analysis tool (Matin spector). The proximal promoter of Inpp5a (from −1419 to −109), which contains 2 GC boxes, was linked to luciferase reporter pGL4.14 (Promega). The reporter assays were performed in both PC12 and HEK293 cells. The PC12 cell lines expressing TBP-13Q or -105Q were transfected with Inpp5a reporter with or without SP1 plasmids. HEK293 cells were transfected with Inpp5a reporter and SP1 with TBP plasmids (TBP-13Q or -105Q). Cells were transfected for 24 h, then collected for luciferase promoter activity assay. Independent experiments were repeated three times. ONE-Glo Luciferase Assay System reagent (Promega, E6120) was used to detect reporter activity using a microplate reader (Synergy H4, BioTek).

**Co-immunoprecipitation and western blot**. Methods for co-immunoprecipitation and western blot were described previously[22]. For in vitro co-immunoprecipitation, HEK293 cells were co-transfected with TBP (TBP-13Q or -105Q) and SP1 plasmids. Cells were harvested with 1% NP40 lysis buffer (150 mM NaCl, 1% NP-40, 2 mM EDTA, 50 mM Tris, pH 8.0, 1 mM phenylmethylsulfonyl fluoride, and protease inhibitor cocktail, Thermo Fisher) 48 h after transfection. Primary antibodies were incubated with 500 μg protein from whole cell lysates overnight at 4 °C. Protein A-beads (20 μl), after pre-blocking, were then added into the mixture and kept for 1 h with rocking at 4 °C. The beads were collected by centrifugation and boiled with 1× sodium dodecyl sulfate (SDS) loading buffer for western blot analysis.

For in vivo co-immunoprecipitation experiments, cerebellum from 1-month-old WT and SCA17 knock-in mice were homogenized in 1% NP40 lysis buffer with protease inhibitor and phosphatase inhibitor cocktails. Approximately 800 μg brain homogenates were incubated with primary antibody overnight at 4 °C. Protein A beads were used to capture the antibody–protein complex.

Protein samples were separated by 4–20% SDS–polyacrylamide gel electrophoresis gel and were then transferred to a polyvinylidene difluoride membrane (Millipore). After blocking, blots were probed with the appropriate antibodies overnight. Western blots were developed using an ECL Prime Chemiluminescence Kit (GE Healthcare) at a Kwik Quant developing and imaging platform.

**Chromatin immunoprecipitation**. ChIP assay with semiquantitative PCR was performed as described previously[44]. Neuro2a cells were transfected with TBP (TBP-13Q or -105Q) and SP1 plasmids. Cells were harvested 48 h after transfection and fixed with 1% formaldehyde for 10 min at 37 °C and transferred to SDS lysis buffer. The cell lysates were sonicated for 6 times, 10 s per time, to shear DNA to lengths between 200 and 1000 bp. ChIP assay was performed according to the protocol of the ChIP Assay Kit (Millipore, 17-295). Anti-SP1 antibody (5 μg, Santa Cruz, sc-17824 X) was used to pull down DNA–protein complex. Primers used to amplify promoter regions are listed as follows: Inpp5a isoform A/B primers, Forward: 5′-CAC AGC TGG CTA TCC AAA CA-3′ and Reverse: 5′-GCT ATG AGT TCG AGG CCA AC-3′; for Inpp5a isoform C primers, Forward: 5′-TCC TGG GTT TTG TGT GTC AA-3′ and Reverse: 5′-CGT AAG CCC CAG GAA TAC AA-3′; PCNA primers, Forward: 5′-TCC TAA GGA TGG AAA CTG CAG CCT-3′ and Reverse: 5′-ATA GGC GAG GGG CAT CAC GG-3′.

**T7 Endonuclease I assay**. T7 Endonuclease I assay was performed as described previously[45]. In brief, genomic DNA was isolated from Neuro2a cells transfected with Cas9 and gRNA plasmids. The target genomic regions were amplified with the following primers: Inpp5a T7 primers, Forward: 5′-AAT CTG TAA AGA ACT TGT GG-3′ and Reverse: 5′-AAC TTC AAT TTG ATC CCT GG-3′. The PCR products were reannealed under the following conditions: 95 °C for 5 min, ramp to 4 °C by 1 °C/30 s, and then incubated with T7 Endonuclease I (New England Bio Labs) for 60 min at 37 °C. The reaction products were subjected to 2% agarose gel electrophoresis.

**IP$_3$ ELISA assay**. Cerebellum of 3-month-old germline SCA17 knock-in mice and age-matched WT mice (three to four mice per genotype) were collected and rinsed in phosphate-buffered saline (PBS). The tissues were finely minced and homogenized in PBS with a glass homogenizer on ice and lysed by ultrasonication, followed by centrifugation at 5000 × g for 5 min. The supernatant was collected for assaying. IP$_3$ ELISA assay was performed following the manufacturer's instruction (Lifespan Biosciences, LS-F25696).

**Statistical analysis**. Data were analyzed using the Prism 8 (GraphPad) software. Two-tailed Student's t test was normally used to compare two groups, and one-way analysis of variance followed with Tukey's multiple comparisons test was

performed for comparison of more than two groups. Statistical tests used are indicated in the figure legends. All quantification data were presented as mean ± SEM. A *P* value of <0.05 was considered statistically significant.

**Reporting summary**. Further information on research design is available in the Nature Research Reporting Summary linked to this article.

## Data availability

Data that support the findings of this study are available from the corresponding author upon reasonable request. The RNA sequencing data are available in NCBI's Gene Expression Omnibus under accession code GSE145067. The source data underlying Figs. 1g, 2d, 3c, 4c, f, e, 5a, d, f–h, and 6b, f–h and Supplementary Figs. 1c, 2d, 3, 4c, d, 5b, and 6c are provided as a Source data file. The full blots are included in the Source data file.

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

## Acknowledgements

We thank Tharp Gregory from Yerkes National Primate Research Center for processing RNA seq raw data and submitting data to NCBI's Gene Expression Omnibus. This work was supported by the National Natural Science Foundation of China grants 81830032, 31872779 (to X.-J.L.), 81701281 (to Q.L.), 2016YFC1306000 (to B.T.), and 81501182 (to Y.P.) and grants from the National Institutes of Health (R01NS095181, NS095279, NS101701, AG019206).

## Author contributions

Q.L. and Y.P. designed the study, performed most experiments, and wrote the manuscript; S.H., P.Y., S.Y., and B.T. provided advice, technical assistance, and support to the study; J.Z., L.J., and S.C. maintained and provided EIIa-Cas9 transgenic mice; Y.P. collected the data and analyzed the data; X.L. and S.L supervised the study and revised the manuscript.

## Competing interests

The authors declare no competing interests.
