## [Peer Review File · Nature Communications]

Reviewers' comments:

Reviewer #1 (Remarks to the Author):

Liu-Q et al claim that SCA17 neuropathology, which is caused by expansions in the transcription factor TBP, affects preferentially the cerebellum due to depletion of the calcium homeostasis factor INPP5A. Furthermore, they claim that overexpression of INPP5A prevents the death of cerebellar Purkinje neurons.

Major criticisms:

(1) The key role of recessive mutations of INPP5A for Purkinje neuron death was previously known, also a neuroprotective effect of INPP5A overexpression on Purkinje neuron number was already reported in SCA2 mouse models (PubMedID 22973002). To show this again for another SCA variant (previously SCA2, now SCA17) provides only limited additional value.

(2) SCA17 is caused by autosomal dominant mutations in the transcription factor TBP. Textbook knowledge says TBP is "a general transcription factor at the core of the DNA-binding multiprotein factor TFIID that binds to the TATA box in a pre-initiation complex that is crucial for the activation of eukaryotic genes transcribed by RNA polymerase II / III / I", so the expression of practically all genes depends on it somehow. Of course INPP5A as protein with highly abundant expression in cerebellum will show higher dependence on it than low-abundance factors, but the mutation affects many thousands of downstream targets. Indeed, the RNAseq studies in the manuscript show that at least 145 specifically down-regulated genes exist in cerebellum, so it is completely unclear why the authors suggest that the pathogenesis is via INPP5A and not via multiple pathways. The authors simply state "the Inpp5a gene was the most interesting candidate". Thus, the title "selective neuropathology ... is mediated by ... INPP5A" as ONE individual factor is a claim that is UNSUBSTANTIATED. The experimental findings speak otherwise, since the toxic aggregation process of any protein in the CNS leads to activation of microglia and deleterious pruning of synapses and neurites, independent of specific neuron projections, according to all literature available.

(3) The authors try to provide evidence for their claim, showing data that the overexpression of INPP5A leads to increased Purkinje neuron calbindin staining, interpreting this observation as a neuroprotective rescue. However, INPP5A is known to reduce IP3 signals, modulate calcium efflux from the endoplasmic reticulum to the cytosol, triggering a changed need to buffer calcium in the cytosol via Ca-binding proteins like Calbindin. This is a functional change within Purkinje neurons which makes them better visible upon immunohistochemistry, so the neuron count may artificially increase, but this is no evidence that more neurons survived.

Minor comments on technical problems:

- oversaturated immunoblots distorted quantification
- coimmunoprecipitation is difficult to quantify and the statistics usually cannot be trusted
- IP3 ELISA assay methods were not found
- qRT-PCR should be quantified by $-\Delta\Delta Ct$ statistics
- which CNS region was used for Nissl staining?

Reviewer #2 (Remarks to the Author):

Nine neurodegenerative diseases are caused by polyglutamine (polyQ) expansions in ubiquitously expressed proteins, and yet in all cases, specific brain regions are susceptible to neurodegeneration. Six of these diseases are spinocerebellar ataxias in which the cerebellum is particularly vulnerable. In this study the authors have identified a mechanism by which the selective vulnerability of the cerebellum (and to a lesser extent the striatum), is caused in SCA17, in which the mutation is in the gene encoding the TATA binding protein (TBP). They have used an elegant combination of genetically modified knock-in mice and in vivo AAV transduction experiments to demonstrate that when mutated, TBP inhibits SP1-mediated induced transcription of *Inpp5a* which encodes a protein (INPP5A) that is highly abundant in the cerebellum. A reduction in INPP5A results in the dysregulation of the 1,4,5-triphosphate (IP3) / Ca²⁺ signalling pathway, resulting in neurodegeneration, and known to be associated with ataxia.

The authors began by expressing mutant TBP in the cerebellum, striatum and prefrontal cortex of wild type mice via AAV stereotaxic injection and showed that the cerebellum was the most vulnerable to polyQ aggregation and neurodegeneration. They also demonstrated that the levels of DARPP32 in the striatum were dramatically reduced. They used RNAseq to identify genes that were dysregulated in these brain regions of 3 month old knock-in SCA17 mice, through which they discovered that the *Inpp5a* gene, which expresses a cerebellar enriched protein, was down-regulated. They show that this gene is regulated by SP1 and that this is inhibited by mutant TBP. Finally, they demonstrate that CRISPR/Cas9-mediated deletion of INPP5A in the cerebellum of wild-type mice led to Purkinje cell degeneration, and INPP5A overexpression decreased IP3 levels and ameliorated Purkinje cell degeneration in SCA17 knock-in mice.

One criticism of this study might be that the authors have not shown a link to patient tissue. However, such studies are not easy, as by the time post-mortem brains are available, the combination of neurodegeneration and inflammation means that changes in gene or protein levels are frequently mis-interpreted. This is an extremely well conducted and well-controlled study and is particularly robust because the authors have used in vivo approaches to dissect the mechanism.

There are some minor comments:

The authors do not say much about the striatal changes that they have identified, which are very interesting. They could make more of this, as striatal degeneration is a feature of Huntington's disease (HD) and SCA17 is the commonest phenocopy of HD, where a genetic diagnosis has been made (Wild and Tabrizi 2007 *Current Opinion Neurology* 6, 681).

Could the authors state which AAV capsid was used. Although this is a virus that they have used previously, and which they have referenced, it would be helpful to the reader to have that information.

Figure 1: the panels for the 105Q cerebellum are not at the same magnification as for all of the other panels in the figure and it would be better if they were consistent. Could the authors also indicate the presence of the inclusions with arrows?

Figure 4 legend: c. should be Venn diagrams

Lines 189 – 191: the sentence is confusing. It would be better to read:

However, isoform C is expressed at a very low level in both the cerebellum and cortex (Supp Fig 4b and c), whereas isoform A is much more abundant.....

Line 338 it should be: TBP is unlikely to be as effective as.....

Gill Bates

Reviewer #3 (Remarks to the Author):

The manuscript by Liu et al. describes novel findings using in vivo models of spinocerebellar ataxia 17 (SCA17) to elucidate pathogenic mechanisms linking mutant TBP interaction with SP1 to pathological downregulation of Inpp5a and selective Purkinje cell (PC) degeneration in this disorder. The authors employed two different types of models, one is AAV-mediated overexpression of mutant TBP with normal and an allelic series of expanded polyQ, and the other is a genetically-accurate Tbp knockin mice with 105Q. Both models appear to show most neurodegeneration in the cerebellum, particularly the PCs, but longer expanded Q in the viral model also elicit degeneration in other brain regions. These findings are supported by RNA-seq studies that showed the largest number of DE genes in the cerebellum compared to striatum and PFC. Among the cerebellum-specific DE genes, the authors focused on Inpp5a, which is highly expressed in PCs, and its LOF has been implicated in ataxia phenotype in mice. They demonstrated an elegant mechanism in which mutant TBP enhanced interaction with Sp1, which is a potent positive regulator of Inpp5a expression at least in cultured cells. Finally, they convincingly showed the crucial role of Inpp5a in PC survival in adult mice. They showed CRISPR/Cas9 knockdown of this gene leads to PC degeneration, while OE of this gene in the TBP-105Q KI mice results in the rescue of PC degeneration and normalization of its metabolic substrate, IP3, a key calcium signaling molecule. Overall, the manuscript is well-written, and the results are novel and compelling, and it represents a major advance in understanding the selective pathogenesis of SCA17.

There are only a few minor concerns.

1. A supplemental table should be included to list all the DE genes between TBP-Q105 and WT mice in the cerebellum, striatum and cortex. Gene set enrichment analyses should be done separately for the different brain regions to identify pathways that are selective to individual brain regions.

2. Line 179. “..P value less than 0.05”. It should be changed to “...corrected P value less than 0.05...”.

3. The authors frequently used human gene names (TBP and INPP5A) to refer to mouse genes. Nomenclature for mouse genes and knockin mice should follow the standard guidelines: Tbp-105Q (for KI mice), Inpp5a for the DE mouse gene. AAV OE should also strictly follow such guidelines.

4. Figure 4g. The quality of the images is poor. The WT image appears to be poorly stitched with a line in the middle, and the KI image seems to be out of focus. They need to be replaced.

Responses to reviewers' comments

We thank the reviewers for the encouraging and constructive comments. We revised the manuscript based on the suggestions provided (the changes are marked in blue in the revised manuscript), and we hope this revision has significantly improved the original manuscript. The detailed responses are shown below (in blue font).

Reviewers' comments:

Reviewer #1 (Remarks to the Author):

Liu-Q et al claim that SCA17 neuropathology, which is caused by expansions in the transcription factor TBP, affects preferentially the cerebellum due to depletion of the calcium homeostasis factor INPP5A. Furthermore, they claim that overexpression of INPP5A prevents the death of cerebellar Purkinje neurons.

Major criticisms:

(1) The key role of recessive mutations of INPP5A for Purkinje neuron death was previously known, also a neuroprotective effect of INPP5A overexpression on Purkinje neuron number was already reported in SCA2 mouse models (PubMedID 22973002). To show this again for another SCA variant (previously SCA2, now SCA17) provides only limited additional value.

We agree that the neuroprotective effect of INPP5A has been reported in a SCA2 mouse model. In that paper, Kasumu et al., reported that mutant but not wild-type Atx2 specifically binds the inositol 1,4,5-trisphosphate receptor (IP₃R) and increases its sensitivity to activation by IP₃. Chronic overexpression of INPP5A, which can inactivate IP₃ in Purkinje cells, could alleviate the Purkinje cell pathology. However, our studies show the selective function of INPP5A in Purkinje cells by comparing SCA17 neuropathology in different brain regions. Further, we provided mechanistic insight into why INPP5A is reduced in SCA17 Purkinje cells. Given that SCA diseases share the common pathological feature in Purkinje cells, our findings highlight the importance of INPP5A for the selective neuropathology in SCAs. We have emphasized this in the revised discussion (line 291-305).

(2) SCA17 is caused by autosomal dominant mutations in the transcription factor TBP. Textbook knowledge says TBP is "a general transcription factor at the core of the DNA-

binding multiprotein factor TFIID that binds to the TATA box in a pre-initiation complex that is crucial for the activation of eukaryotic genes transcribed by RNA polymerase II / III / I", so the expression of practically all genes depends on it somehow. Of course, INPPA as protein with highly abundant expression in cerebellum will show higher dependence on it than low-abundance factors, but the mutation affects many thousands of downstream targets. Indeed, the RNAseq studies in the manuscript show that at least 145 specifically down-regulated genes exist in cerebellum, so it is completely unclear why the authors suggest that the pathogenesis is via INPP5A and not via multiple pathways. The authors simply state "the Inpp5a gene was the most interesting candidate". Thus, the title "selective neuropathology ... is mediated by ... INPP5A" as ONE individual factor is a claim that is UNSUBSTANTIATED. The experimental findings speak otherwise, since the toxic aggregation process of any protein in the CNS leads to activation of microglia and deleterious pruning of synapses and neurites, independent of specific neuron projections, according to all literature available.

We appreciate the reviewer's critical comments. TBP is an important transcription factor, however, polyQ expansion in TBP does not affect many thousands of downstream targets. In our previous study, microarray on the cerebella from SCA17 transgenic mice only identified a relatively small number of genes (about 400 genes) that show differential expression (Friedman et al et al. 2007). In this study, RNA sequencing revealed only 245 cerebellar specifically dysregulated genes in SCA17 KI mice. This is because mutant TBP is still functional as shown in our previous study (Friedman et al. 2007, 2008) and does not cause the same consequences as loss of TBP, which would affect many downstream targets. All these highlight the importance of studying specifically dysregulated genes and the association with selective Purkinje cell neuropathology in SCA17.

RNA sequencing identified 245 cerebellar specifically dysregulated genes and 110 up-regulated genes that were enriched in immune related pathways, and no gene is exclusively expressed highly in cerebellum (Supplementary table 1). Therefore, we focused on the 145 specifically down-regulated genes (Supplementary table 1). All 145 genes were ranked based on the following criteria: 1) abundant expression in the cerebellum; 2) relative highly expressed in the cerebellum (expression in the cerebellum is at least three times higher than that in the striatum and prefrontal cortex (supplementary table 1). Based on the above screen criteria and current literature, *Inpp5a* is the most interesting candidate gene. We have included these in the revised text (line 154-165).

In this study, INPP5A was identified as an important molecule for the selective Purkinje cell degeneration. Of course, the alterations of multiple genes were also seen in the cerebellum of SCA17 KI mice. However, whether these genes are involved in SCA17 remains to be verified functionally. We agree with the reviewer that other factors could also be involved in selective neuropathology in SCA17. Thus, we have changed the title to **“The cerebellum-enriched protein INPP5A is involved in selective neuropathology in Spinocerebellar ataxias 17 mice”**.

(3) The authors try to provide evidence for their claim, showing data that the overexpression of INPP5A leads to increased Purkinje neuron calbindin staining, interpreting this observation as a neuroprotective rescue. However, INPP5A is known to reduce IP3 signals, modulate calcium efflux from the endoplasmic reticulum to the cytosol, triggering a changed need to buffer calcium in the cytosol via Ca-binding proteins like Calbindin. This is a functional change within Purkinje neurons which makes them better visible upon immunohistochemistry, so the neuron count may artificially increase, but this is no evidence that more neurons survived.

We thank the reviewer for this point. Immunostaining of Purkinje cells with anti-calbindin has been a standard assay to detect the degeneration of Purkinje cells (Park et al. 2013, Rousseaux et al. 2018). The reviewer thought that overexpression of INPP5A may alter calcium in the cytosol and make Purkinje neurons more visible so that the increased calbindin staining after INPP5A overexpression does not suggest that more neurons were survived. To provide more evidences, we included new Nissl staining results on the cerebellum, which also showed increased Purkinje cells after INPP5A overexpression (supplementary Fig. 6c). Combined with the WB results that also showed altered expression of calbindin (Fig.6f, g), we think we provide good evidence for the protective effect of INPP5A to increase the number of Purkinje cells.

Minor comments on technical problems:

- oversaturated immunoblots distorted quantification

We thank the reviewer for pointing this out. We have replaced the oversaturated immunoblots with less exposed blots that come from the same blots in the revised figures (Fig3c, d, Fig4f, Fig5c, e and Fig6b, f).

- coimmunoprecipitation is difficult to quantify and the statistics usually cannot be trusted

We chose the most representative images in the figures. We repeated the IP experiment several times and quantified the average value.

- IP3 ELISA assay methods were not found

We thank the reviewer for pointing this out. We have added IP3 ELISA assay method in the revised text (line 482-487).

- qRT-PCR should be quantified by -Delta Delta-Ct statistics

We agree with the reviewer's comment. We used the comparative CT method to quantify relative gene expression, which is also referred to as the $2^{-\Delta\Delta CT}$ method (Schmittgen and Livak 2008). We have mentioned this in the revised text (line 402).

- which CNS region was used for Nissl staining?

Nissl staining results (supplementary fig.1b) was performed on the cerebellum. We have added a subtitle for supplementary fig. 1b and 1c.

Reviewer #2 (Remarks to the Author):

Nine neurodegenerative diseases are caused by polyglutamine (polyQ) expansions in ubiquitously expressed proteins, and yet in all cases, specific brain regions are susceptible to neurodegeneration. Six of these diseases are spinocerebellar ataxias in which the cerebellum is particularly vulnerable. In this study the authors have identified a mechanism by which the selective vulnerability of the cerebellum (and to a lesser extent the striatum), is caused in SCA17, in which the mutation is in the gene encoding the TATA binding protein (TBP). They have used an elegant combination of genetically modified knock-in mice and in vivo AAV transduction experiments to demonstrate that when mutated, TBP inhibits SP1-mediated induced transcription of Inpp5a which encodes a protein (INPP5A) that is highly abundant in the cerebellum. A reduction in INPP5A results in the dysregulation of the 1,4,5-triphosphate (IP3) / Ca²⁺ signalling pathway, resulting in neurodegeneration, and known to be associated with ataxia.

The authors began by expressing mutant TBP in the cerebellum, striatum and prefrontal cortex of wild type mice via AAV stereotaxic injection and showed that the cerebellum was the most vulnerable to polyQ aggregation and neurodegeneration. They also demonstrated that the levels of DARPP32 in the striatum were dramatically reduced. They used RNAseq to identify genes that were dysregulated in these brain regions of 3 month old knock-in SCA17 mice, through which they discovered that the *Inpp5a* gene, which expresses a cerebellar enriched protein, was down-regulated. They show that this gene is regulated by SP1 and that this is inhibited by mutant TBP. Finally, they demonstrate that CRISPR/Cas9-mediated deletion of *INPP5A* in the cerebellum of wild-type mice led to Purkinje cell degeneration, and *INPP5A* overexpression decreased IP3 levels and ameliorated Purkinje cell degeneration in SCA17 knock-in mice.

One criticism of this study might be that the authors have not shown a link to patient tissue. However, such studies are not easy, as by the time post-mortem brains are available, the combination of neurodegeneration and inflammation means that changes in gene or protein levels are frequently mis-interpreted. This is an extremely well conducted and well-controlled study and is particularly robust because the authors have used in vivo approaches to dissect the mechanism.

We appreciate the reviewer's positive and encouraging comments. As the reviewer said, SCA17 is a rare neurodegenerative disorder. It is hard to obtain SCA17 brain tissues.

There are some minor comments:

The authors do not say much about the striatal changes that they have identified, which are very interesting. They could make more of this, as striatal degeneration is a feature of Huntington's disease (HD) and SCA17 is the commonest phenocopy of HD, where a genetic diagnosis has been made (Wild and Tabrizi 2007 Current Opinion Neurology 6, 681).

This is an interesting perspective and we appreciate the reviewer's comments. Since the major pathology of SCA17 is cerebellar atrophy and selective loss of Purkinje cells, we did not focus on the striatal changes in this study. In the revised text, however, we supply more immunostaining of DARPP32 in striatum (Fig. 3e) and found the number of DARPP32-positive cells was slightly reduced. We did gene set enrichment analysis on striatum-specific differentiated genes and found that most upregulated genes are immune

related pathway and most downregulated genes involved in calcium transport or calcium transmembrane transport (supplementary Fig. 3d and 3k). We checked literature and did gene function annotations one by one for some of striatal enriched downregulated genes. We found some of them are involved in dopamine signaling (such as *Strn*, *Arpp21*) and some of them are associated with striatal degeneration, mental retardation, dyskinesia, ataxia and epilepsy (such as *Camk2b*, *Pde10a*, *Kcnab1*, *Tomm70a* and *Cacna2d3*) (supplementary Tab. 1). We think all the evidence supports the striatum degeneration in SCA17 and may provide some clues for why SCA17 shares some pathological features with HD. Thus, we have included these new results in the revised text (line 316-327).

Could the authors state which AAV capsid was used? Although this is a virus that they have used previously, and which they have referenced, it would be helpful to the reader to have that information.

All AAVs were serotype 9. We have included this information in the revised text.

Figure 1: the panels for the 105Q cerebellum are not at the same magnification as for all of the other panels in the figure and it would be better if they were consistent. Could the authors also indicate the presence of the inclusions with arrows?

We have indicated the inclusions with arrows in the revised figures (Fig.1d, e). Actually, the magnification for all the panels are the same. Overexpressed mutant TBP with 105Q is readily to form aggregates of varying sizes in the nucleus. When comparing with the size of the nuclei by DAPI staining, their size can then be estimated.

Figure 4 legend: c. should be Venn diagrams

We thank the reviewer for pointing this out and have corrected this in the revised figure legend.

Lines 189 – 191: the sentence is confusing. It would be better to read: However, isoform C is expressed at a very low level in both the cerebellum and cortex (Supp Fig 4b and c), whereas isoform A is much more abundant...

We thank the reviewer for the above suggestion and have corrected this sentence in the revised text.

Line 338 it should be: TBP is unlikely to be as effective as...

We have made the above change in the revised text.

Reviewer #3 (Remarks to the Author):

The manuscript by Liu et al. describes novel findings using in vivo models of spinocerebellar ataxia 17 (SCA17) to elucidate pathogenic mechanisms linking mutant TBP interaction with SP1 to pathological downregulation of Inpp5a and selective Purkinje cell (PC) degeneration in this disorder. The authors employed two different types of models, one is AAV-mediated overexpression of mutant TBP with normal and an allelic series of expanded polyQ, and the other is a genetically-accurate Tbp knockin mice with 105Q. Both models appear to show most neurodegeneration in the cerebellum, particularly the PCs, but longer expanded Q in the viral model also elicit degeneration in other brain regions. These findings are supported by RNA-seq studies that showed the largest number of DE genes in the cerebellum compared to striatum and PFC. Among the cerebellum-specific DE genes, the authors focused on Inpp5a, which is highly expressed in PCs, and its LOF has been implicated in ataxia phenotype in mice. They demonstrated an elegant mechanism in which mutant TBP enhanced interaction with Sp1, which is a potent positive regulator of Inpp5a expression at least in cultured cells. Finally, they convincingly showed the crucial role of Inpp5a in PC survival in adult mice. They showed CRISPR/Cas9 knockdown of this gene leads to PC degeneration, while OE of this gene in the TBP-105Q KI mice results in the rescue of PC degeneration and normalization of its metabolic substrate, IP3, a key calcium signaling molecule. Overall, the manuscript is well-written, and the results are novel and compelling, and it represents a major advance in understanding the selective pathogenesis of SCA17.

There are only a few minor concerns.

1. A supplemental table should be included to list all the DE genes between TBP-Q105 and WT mice in the cerebellum, striatum and cortex. Gene set enrichment analyses should be done separately for the different brain regions to identify pathways that are selective to individual brain regions.

We appreciate the reviewer's helpful comments. We have included a new supplementary table 1 that contains the differentially expressed genes in the cerebellum, striatum and

prefrontal cortex. We also performed Gene enrichment analyses for these differentially expressed genes in the CB/STR/PFC separately (supplementary fig. 3) and found some interesting results. In the cerebellum, cerebellum-specific down-regulated genes were found to be involved in cellular processes like synaptic transmission and signaling, whereas most striatum-specific down-regulated genes are involved in calcium transport or calcium transmembrane transport. We have included gene set enrichment analyses in method and discussion (line 316-327).

2. Line 179. "...*P* value less than 0.05". It should be changed to "...corrected *P* value less than 0.05...".

We followed the reviewer's suggestion and made this change in the revised text.

3. The authors frequently used human gene names (*TBP* and *INPP5A*) to refer to mouse genes. Nomenclature for mouse genes and knockin mice should follow the standard guidelines: *Tbp-105Q* (for *KI* mice), *Inpp5a* for the *DE* mouse gene. *AAV OE* should also strictly follow such guidelines.

We thank the reviewer for pointing this out and have corrected the gene names in the revised text.

4. Figure 4g. The quality of the images is poor. The *WT* image appears to be poorly stitched with a line in the middle, and the *KI* image seems to be out of focus. They need to be replaced.

We thank the reviewer for pointing this out. We have used new and focused images in the revision.

References

Friedman, M. J., A. G. Shah, Z. H. Fang, E. G. Ward, S. T. Warren, S. Li and X. J. Li (2007). "Polyglutamine domain modulates the TBP-TFIIB interaction: implications for its normal function and neurodegeneration." *Nat Neurosci* 10(12): 1519-1528.

Friedman, M. J., C. E. Wang, X. J. Li and S. Li (2008). "Polyglutamine expansion reduces the association of TATA-binding protein with DNA and induces DNA binding-independent neurotoxicity." *J Biol Chem* 283(13): 8283-8290.

Park, J., I. Al-Ramahi, Q. Tan, N. Mollema, J. R. Diaz-Garcia, T. Gallego-Flores, H. C. Lu, S. Lagalwar, L. Duvick, H. Kang, Y. Lee, P. Jafar-Nejad, L. S. Sayegh, R. Richman, X. Liu, Y. Gao, C. A. Shaw, J. S. C. Arthur, H. T. Orr, T. F. Westbrook, J. Botas and H. Y. Zoghbi (2013). "RAS-MAPK-MSK1 pathway modulates ataxin 1 protein levels and toxicity in SCA1." *Nature* 498(7454): 325-331.

Rousseaux, M. W. C., T. Tschumperlin, H. C. Lu, E. P. Lackey, V. V. Bondar, Y. W. Wan, Q. Tan, C. J. Adamski, J. Friedrich, K. Twaroski, W. Chen, J. Tolar, C. Henzler, A. Sharma, A. Bajic, T. Lin, L. Duvick, Z. Liu, R. V. Sillitoe, H. Y. Zoghbi and H. T. Orr (2018). "ATXN1-CIC Complex Is the Primary Driver of Cerebellar Pathology in Spinocerebellar Ataxia Type 1 through a Gain-of-Function Mechanism." *Neuron* 97(6): 1235-1243 e1235.

Schmittgen, T. D. and K. J. Livak (2008). "Analyzing real-time PCR data by the comparative C(T) method." *Nat Protoc* 3(6): 1101-1108.

REVIEWERS' COMMENTS:

Reviewer #2 (Remarks to the Author):

In their response to the reviewers comments and the revisions that have been made to the manuscript, the authors have satisfied the concerns of this referee.

I have also looked through their response to referee 1 and the corrections that have been made to the manuscript to answer the issues that reviewer 1 has raised. I think that all of the points have been addressed either by the addition of new text or modification of the figures.

Reviewer #3 (Remarks to the Author):

The authors have fully addressed all the concerns raised by this reviewer., and the study is appropriate for publication now.

Responses to reviewers' comments

REVIEWERS' COMMENTS:

Reviewer #2 (Remarks to the Author):

In their response to the reviewers comments and the revisions that have been made to the manuscript, the authors have satisfied the concerns of this referee.

I have also looked through their response to referee 1 and the corrections that have been made to the manuscript to answer the issues that reviewer 1 has raised. I think that all of the points have been addressed either by the addition of new text or modification of the figures.

We thank the reviewer for the positive comments.

Reviewer #3 (Remarks to the Author):

The authors have fully addressed all the concerns raised by this reviewer., and the study is appropriate for publication now.

We thank the reviewer for the positive comments.